# Hypoxia with or without Treadmill Exercises Affects Slow-Twitch Muscle Atrophy and Joint Destruction in a Rat Model of Rheumatoid Arthritis

**DOI:** 10.3390/ijms24119761

**Published:** 2023-06-05

**Authors:** Yoichiro Kamada, Yuji Arai, Shogo Toyama, Atsuo Inoue, Shuji Nakagawa, Yuta Fujii, Kenta Kaihara, Ryota Cha, Osam Mazda, Kenji Takahashi

**Affiliations:** 1Department of Orthopaedics, Graduate School of Medical Science, Kyoto Prefectural University of Medicine, Kawaramachi-Hirokoji, Kamigyo-ku, Kyoto 602-8566, Japan; 2Department of Sports and Para-Sports Medicine, Graduate School of Medical Science, Kyoto Prefectural University of Medicine, Kawaramachi-Hirokoji, Kamigyo-ku, Kyoto 602-8566, Japan; 3Department of Immunology, Graduate School of Medical Science, Kyoto Prefectural University of Medicine, Kawaramachi-Hirokoji, Kamigyo-ku, Kyoto 602-8566, Japan

**Keywords:** rheumatoid arthritis, hypoxia, treadmill running, slow-twitch muscle

## Abstract

The effects of treadmill running under hypoxic conditions on joints and muscles of collagen-induced arthritis (CIA) rats were investigated. CIA rats were divided into normoxia no-exercise, hypoxia no-exercise (Hypo-no), and hypoxia exercise (Hypo-ex) groups. Changes were examined on days 2 and 44 of hypoxia with or without treadmill exercises. In the early stage of hypoxia, the expression of hypoxia-inducible factor (HIF)-1α increased in the Hypo-no and Hypo-ex groups. The expression of the egl-9 family hypoxia-inducible factor 1 (EGLN1) and vascular endothelial growth factor (VEGF) in the Hypo-ex group also increased. Under sustained hypoxia, the Hypo-no and Hypo-ex groups did not show increased expression of HIF-1α or VEGF, but p70S6K levels were elevated. Histologically, joint destruction was alleviated in the Hypo-no group, the loss of muscle weight in slow-twitch muscles was prevented, and muscle fibrosis was suppressed. In the Hypo-ex group, the preventive effect of a reduction in the slow-twitch muscle cross-sectional area was enhanced. Thus, chronic hypoxia in an animal model of rheumatoid arthritis controlled arthritis and joint destruction and prevented slow-twitch muscle atrophy and fibrosis. The combination of hypoxia with treadmill running further enhanced the preventive effects on slow-twitch muscle atrophy.

## 1. Introduction

Rheumatoid arthritis (RA) is an autoimmune disease that primarily affects the synovial membranes and causes excessive production of pro-inflammatory cytokines and chemokines, such as tumor necrosis factor (TNF)-α and interleukin (IL)-6 in synovial membranes. This leads to joint destruction, pain, and a limited range of motion [1,2]. RA causes inflammation not only in the joints but also in the muscles themselves, resulting in muscle fibrosis and muscle atrophy [3]. In an animal model of RA, Kamada et al. found predominantly slow-twitch muscle fiber atrophy [4]. RA affects the entire body, and because the loss of muscle mass in the entire body decreases the activities of daily living (ADL), muscle mass is an important aspect in the evaluation of RA progression and treatment success.

RA treatment has focused on pharmacotherapy. The advent of biological disease-modifying anti-rheumatic drugs (bDMARDs) around the year 2000 has brought about a paradigm shift in the pharmacotherapy of RA, making it possible to minimize joint destruction [5,6,7]. However, the availability of bDMARDs is limited due to adverse events, such as infection and allergy, as well as drug costs [8,9]. On the other hand, exercise therapy is a safe and economical treatment method and is widely used for the treatment of systemic disorders, as well as joint diseases. Until recently, exercise therapy has been based on the experience of individual physicians and therapists. However, the mechanisms of exercise therapy are now being elucidated, and evidence is accumulated. Exercise therapy is strongly recommended for RA in Cochrane reviews [10,11] and has been reported to clinically improve not only joint symptoms but also ADL and physical function as measured by muscle strength and cardiopulmonary function [12,13]. It has also been reported that exercise therapy using a treadmill has a protective effect on articular cartilage and subchondral bones [14]. Treadmill running in a rat model of RA inhibits the production of Cx43 and TNF-α in the synovial membrane, as well as the degeneration of articular cartilage and subchondral bones [15], and it prevents the atrophy of slow-twitch muscles [4]. These findings indicate that exercise therapy inhibits joint destruction and muscle atrophy via the regulation of pro-inflammatory cytokine levels.

Moreover, a hypoxic environment induces the expression of hypoxia-inducible factor (HIF)-1α in cartilage, which protects cartilage [16]. Shimomura et al. reported that HIF-1α regulates aggrecan and ADAM-TS5 expression in response to mechanical stimulation under hypoxia [17]. In addition, hypoxia in muscles promotes muscle hypertrophy via HIF-1α expression [18]. Furthermore, Kaihara et al. reported that a sustained hypoxic environment suppresses synovitis and controls joint destruction in an animal model of RA [19], indicating that synovitis caused by RA can be suppressed by hypoxic therapy.

Based on the above background, we hypothesized that combining a hypoxic environment with exercise therapy would have additional effects on preventing joint destruction and muscle atrophy in rat models of RA. The purpose of this study was to examine the effects of rearing and treadmill running in a hypoxic environment on joint destruction and slow-twitch muscle atrophy in a rat model of RA.

## 2. Results

### 2.1. Changes in Body Weight and Paw Volume over Time

The body weight (BW) increased slowly from day 0 to day 10 in each study group, decreased from day 12 to around day 20, and then increased again (Figure 1a). On day 38 and day 40, BW values in the Hypo-no group were significantly reduced compared to those in the Hypo-ex group (*p* < 0.05). The paw volume increased from around day 12 in each group, reached its maximum on days 18–20, and then decreased slowly (Figure 1b). The paw volumes in the Hypo-no and Hypo-ex groups on day 14 and day 16 were lower than the Normo-no group (*p* < 0.05). The clinical score of the Normo-no group on day 14 was higher than the Hypo-no and Hypo-ex groups (*p* < 0.05). Moreover, the clinical score of the Normo-no group on day 16 was higher than the Hypo-no group (*p* < 0.05; Figure 1c).

### 2.2. The Effects of Treadmill Running in a Hypoxic Environment on Articular Cartilages

To evaluate the histological effects on the joints, rat ankle cartilage samples on day 44 were stained with safranin O. In the Normo-no group, irregularly shaped articular cartilages, cartilage thinning, and decreased safranin O staining were observed, and the cartilage destruction scores in this group were higher than the Hypo-no and Hypo-ex groups (*p* < 0.05; Figure 2a,b).

### 2.3. The Effects of Treadmill Running in a Hypoxic Environment on Skeletal Muscles

#### 2.3.1. Muscle Weight on Day 44

No significant difference in the relative extensor digitorum longus (EDL) weight was found among the groups (Figure 3a). However, the relative soleus weight values in the Hypo-no and Hypo-ex groups were 1.23- and 1.36-fold higher than the Normo-no group (*p* < 0.05 and *p* < 0.01, respectively; Figure 3b).

#### 2.3.2. Histological Evaluation of the Soleus Muscle on Day 44

To histologically evaluate the muscles of each group, we performed laminin and picrosirius-red staining of the soleus muscle (Figure 4a,b). The relative muscle cross-sectional area of the soleus in the Hypo-ex group was significantly increased compared to the Normo-no and Hypo-no groups (1.52- and 1.24-fold, *p* < 0.01 and *p* < 0.05, respectively; Figure 4b). The fibrosis rates of the soleus muscles in the Hypo-no and Hypo-ex groups were 22% and 40% less than the Normo-no group (*p* < 0.05 and *p* < 0.01, respectively; Figure 4c). 

#### 2.3.3. Gene Expression on Days 2 and 44

On day 2, the HIF-1α expression levels in the soleus muscles of the Hypo-no and Hypo-ex groups were 2.43- and 3.15-fold higher than the Normo-no group (*p* < 0.01; Figure 5a). Likewise, the expression levels of vascular endothelial growth factor (VEGF) in the Hypo-ex group were elevated compared to the Normo-no and Hypo-no groups (6.35- and 2.19-fold, respectively, *p* < 0.01; Figure 5b), as were the expression levels of the egl-9 family hypoxia-inducible factor 1 (EGLN1) in the Hypo-ex group compared to the Normo-no and Hypo-no groups (2.22- and 1.47-fold, *p* < 0.01 and *p* < 0.05, respectively; Figure 5b,c). By contrast, no significant differences in soleus expression levels of HIF-1α, VEGF, and atrogin-1, a marker of muscle protein degradation, among experimental groups were observed on day 44 (Figure 6a–c,e). However, the expression of p70S6K, a marker of muscle protein synthesis, was significantly higher in the Hypo-no (1.49-fold) and Hypo-ex (1.46-fold) groups than the Normo-no group (*p* < 0.05 and *p* < 0.01, respectively; Figure 6d). 

## 3. Discussion

The most important finding of this study in rats with collagen-induced arthritis (CIA) is that the combination of treadmill running in a sustained hypoxic environment was effective not only in preventing joint destruction but also slow muscle atrophy and fibrosis. 

Kaihara et al. reported no effect on body weight after rearing CIA rats for 28 days in a sustained hypoxic environment of 12% O_2_ [19]. In the present study, CIA rats were reared in a continuously hypoxic environment of 12% O_2_ for 42 days with or without treadmill exercise in the last 14 days. There was no difference in body weight between the groups most of the time, suggesting that rearing and exercise therapy under these conditions have few, minor adverse effects on the whole body. 

Oxygen levels are reduced in RA joints compared to healthy joints [20], and HIF-1α expression is abnormally activated in the synovium. Downstream pro-inflammatory cytokines such as TNF-α, IL-6, and IL-1β are activated, and joint destruction progresses by promoting angiogenesis, pannus formation, and inflammatory processes [21]. Therefore, placing RA patients in a hypoxic environment may further increase HIF-1α expression and promote joint destruction. However, Kaihara et al. reported that a sustained hypoxic environment transiently increases the protein expression of HIF-1α in RA synovial cell lines, but then gradually decreases its expression and suppresses the expression of inflammatory cytokines [19]. In a comparison of 28-day normoxic and 12% sustained hypoxic CIA rats, these authors reported that in the sustained hypoxic group, joint swelling was suppressed, and joint destruction was controlled 2–3 weeks after immune sensitization. The results also showed that the expression of HIF-1α was decreased via the activation of the prolyl hydroxylase domain (PHD), and joint destruction was suppressed due to persistent hypoxia. When comparing, in the present study, normoxic and chronic hypoxic rats reared for 42 days, both the paw volume and clinical score of the Hypo-no group were decreased compared to the Normo-no group on days 14 and 16, i.e., at the time when inflammatory processes were strongest. Histologically, cartilage destruction was alleviated in the Hypo-no group compared to the Normo-no group at the end of the study period on day 44. HIF-1α expression in the synovium decreased even after long-term rearing in a sustained hypoxic environment in comparison to levels following 28-day hypoxia, as reported by Kaihara et al., and this mechanism suppressed the joint swelling and destruction in CIA rats. On the other hand, exercise induces various forms of mechanical stress on articular cartilage and the synovium. Bartalucci et al. reported that the effects of training are dependent on complex, adaptive changes that are induced by acute physical exercise at different levels. In particular, the hypothalamus–pituitary–adrenocortical axis, as well as the sympatho-adrenomedullary system, are mainly involved in mediating the physiological effects of physical exercise [22]. Toti et al. reported that high-intensity exercise, in addition to metabolic changes consisting of a decrease in blood lactate and body weight, induces an increase in mitochondrial enzyme levels and slow fiber numbers in different skeletal muscles of mice, which indicates an exercise-induced increase in aerobic metabolism [23]. Excessive mechanical stress on articular cartilage is a risk for osteoarthritis, whereas moderate mechanical stress is anabolic [24,25]. González-Chávez et al. reported that low-intensity physical exercise decreases joint damage and the expression of RA-related genes and signaling pathways [26]. Shimomura et al. reported that HIF-1α regulates aggrecan and ADAM-TS5 expression in response to mechanical stimulation under hypoxia and general mechanical stimulation in articular cartilage under hypoxia while controlling cartilage homeostasis [17]. In addition, Shimomura et al. reported that moderate-intensity treadmill exercise suppresses the production of inflammatory cytokines, such as TNF-α, via the downregulation of connexins in the synovial membrane, as well as joint destruction, in an RA rat model [15]. In the present study, the combined effects of 14-day moderate-intensity treadmill running and 42-day hypoxia on the suppression of joint swelling and destruction in CIA rats were investigated. The results indicate that the combination of sustained hypoxia and moderate-intensity treadmill exercise has the same inhibitory effect on joint destruction as sustained hypoxia alone. The lack of significant differences between the Hypo-no and Hypo-ex groups suggests that the suppression of both HIF-1α expression in the synovium and connexin expression does not enhance the TNF-α-suppressive effect in CIA rats. 

Hypoxic environments and HIF-1α expression in muscles have been reported to promote muscle hypertrophy. When blood flow is restricted at the thigh base of athletes, an increase in blood IL-6 levels leading to muscle hypertrophy is observed [27]. Thermal coagulation of the femoral vein in rats causes muscle hypertrophy [28], and the phosphorylation of p70S6K, which leads to muscle hypertrophy, is increased when blood circulation is restricted in rats [29]. In vitro, HIF-1α is required for the induction and differentiation of the C2C12 cell line into skeletal muscle fibers, HIF-1α knockout inhibits this differentiation, and myotubes differentiating from the C2C12 cell line become hypertrophic upon ischemic preconditioning. Elevated HIF-1α expression leads to muscle hypertrophy via the nonclassical Wnt pathway [30]. We have previously reported that treadmill exercise prevents slow-twitch muscle atrophy in an animal model of RA [4]. In the present study, the Hypo-ex and Hypo-no groups had increased soleus muscle weight and decreased muscle fibrosis compared to the Normo-no group on day 44. The cross-sectional area of the soleus muscle was significantly larger in the Hypo-ex group, indicating that soleus muscle atrophy can be prevented in a hypoxic environment and that the combination with treadmill exercise is highly effective in this regard. We investigated the mechanism of this phenomenon and found that the expression levels of HIF-1α, EGLN1, and VEGF in the Hypo-ex group were upregulated compared to the Normo-no group in the early stage of hypoxia, whereas the EGLN1 and VEGF levels were higher in the Hypo-ex group than the Normo-no group. HIF-1α influenced the atrophy of the soleus muscles in CIA rats in the early hypoxic period. However, under sustained hypoxia, HIF-1α is degraded, and its expression in muscle is reduced due to increased PHD activity with negative feedback. In the present study, elevated mRNA expression of hypoxia-related factors was suppressed on day 44 and did not differ significantly between groups. The p70S6K expression level was elevated only in the Hypo-ex group. The oxygen concentration in muscles during exercise is estimated to be 1/10 of that at rest, and exercise induces hypoxia in muscles, leading to HIF-1α expression [31]. In the present study, HIF-1α expression, which was initially decreased in the Hypo-ex group, had been restored by the time the treadmill exercise started on day 28, which increased p70S6K levels and had a stimulatory effect on muscle synthesis. The inhibitory effect of sustained hypoxia on joint destruction may also have effectively counteracted soleus muscle atrophy by improving motor function of the lower extremities. 

There are several limitations to this study. For instance, the expression of HIF-1α and other hypoxia-related factors can change over time, but we did not analyze the expression at all time points. Experiments with pharmacological or genetic target inhibition to investigate the involved pathways have not been performed. Moreover, low-intensity exercise or other exercise conditions may have a more positive effect on articular cartilage and slow-twitch muscles. Therefore, details of the underlying mechanisms may remain unclear. 

## 4. Materials and Methods

### 4.1. Collagen-Induced Arthritis Model

The CIA rat model has several similarities with human RA and has been widely used for in vivo RA studies. To create this model for our study, type II collagen (Collagen Research Center, Tokyo, Japan) and Freund’s incomplete adjuvant (Sigma-Aldrich, St. Louis, MO, USA) were mixed and emulsified in a 1:1 ratio on ice. This solution (200 μL) was injected intradermally into the base of the tail of 8-week-old male Dark Agouti (DA) rats (body weight, 150–165 g; Shimizu Laboratory Suppliers, Kyoto, Japan). Rats were kept in a 12 h light/dark cycle with free access to food and water.

### 4.2. Body Weight, Paw Volume, and Clinical Score

Body weight, paw volume, and clinical score were measured on days 0, 2, 4, 6, 8, and 10 post-immunization and daily after day 12. The paw volume was measured using a water replacement plethysmometer (Unicom Japan, Tokyo, Japan). 

The clinical score was defined as follows: score 0, normal paw; score 1, inflammation and swelling of one toe; score 2, inflammation and swelling of more than one toe with inflammation and swelling of the entire paw or mild swelling of the entire paw; score 3, inflammation and swelling of the entire paw; and score 4, severe inflammation and swelling of the entire paw or ankylosed paw [32]. 

### 4.3. Evaluation of the EDL and Soleus Muscle Weights

The EDL and soleus muscles from both legs were collected and weighed on day 44. Muscle mass is presented as muscle mass/BW ratio. 

### 4.4. Rearing in the Hypoxic Chamber and Treadmill Running

We developed a unique hypoxia chamber (Natsume Seisakusho Co., Ltd., Wakenyaku Co., Ltd., Kyoto, Japan) in which the oxygen concentration can be arbitrarily set (Figure 7). Nitrogen generated by an N_2_ gas generator is mixed with ambient air at an arbitrary ratio using an N_2_^+^ air blender to induce hypoxia. The hypoxic air circulates through the breeding and workplace cages to create a hypoxic environment in the chamber. The oxygen concentration can be measured at any location using a gas analyzer. The oxygen concentration in the chamber ranged from 4% to 20%. The oxygen concentration conditions were examined in preliminary experiments, and 12% O_2_ was set as the concentration that had the least effect on the body weight and other systemic conditions of healthy DA rats. Rats can be kept in this chamber, and treadmill running can be performed seamlessly. 

### 4.5. Treadmill Running Protocol

Eight-week-old male DA rats were randomly divided into the following three groups (N = 6 each): CIA rat normoxia-sedentary group (Normo-no), CIA rat hypoxia-sedentary group (Hypo-no), and CIA rat hypoxia-treadmill running group (Hypo-ex). The protocol for each group is presented in Figure 8b. Only the Hypo-ex group was forced to run from day 28 to day 42 on a treadmill device (TMS8D; MEQUEST, Toyama, Japan). The running protocol used for this group had previously been reported to have an inhibitory effect on muscle atrophy in the rat RA model and was as follows: running 5 times/week, 12 m/min, and 30 min/day [4]. As an additional experiment, CIA rats were divided into three groups and performed a single treadmill exercise on day 2 to examine the short-term effects of treadmill running in a hypoxic environment (Figure 8a). To analyze how muscle atrophy and fibrosis are affected by a single or repeated bout of exercise in a short time and a sustained hypoxic environment, rats were euthanized either 2 h after the first treadmill exercise on day 2 (N = 4) or 48 h after the last treadmill exercise on day 44 (N = 6).

### 4.6. Histochemical Analysis (Articular Cartilage)

Two days after the end of the treadmill running protocol, the right ankle joints of the rats were removed and kept in 4% paraformaldehyde (Wako, Osaka, Japan). They were then decalcified with 20% ethylenediaminetetraacetic acid and embedded in paraffin. The center of each ankle joint was sliced into 6 µm-thick sagittal slices and stained with safranin O.

Cartilage destruction was measured on a scale from 0 to 3, from no damage to completely destroyed cartilage layers, as described in Weinberger et al. [33].

### 4.7. Muscle Preparation

After euthanizing the rats on day 2 or day 44, each rat was weighed individually. The EDL and soleus muscles from both legs were collected and weighed. Excess fat and connective tissue were removed using standardized dissection methods. The soleus muscles excised from one lower limb were fixed in 4% paraformaldehyde (Wako) for histological studies. The soleus of the other lower limb was immersed in RNA Protect^®^ Tissue Reagent (QIAGEN, Hilden, Germany) for genetic analysis and stored at 4 °C until further analysis.

### 4.8. Muscle Histochemistry

Each muscle was paraffin-embedded, and 10 μm-thick sections were prepared from the center of the muscle using a cryostat. Picrosirius red staining was performed using the following protocol. The slides were deparaffinized and hydrated. The slides were rinsed with distilled water, and sections were stained with Weigert’s iron hematoxylin for 5 min. Then, the slides were washed in running tap water and rinsed with distilled water. The sections were stained in picrosirius red for 20 min. After the sections were dehydrated with isopropyl alcohol, they were cleared and mounted. Laminin staining was using the following protocol. The slides were deparaffinized and hydrated. The slides were rinsed with distilled water and treated with proteinase K for 8 min at room temperature. The slides were treated with 3% hydrogen peroxide water and reacted with anti-laminin rabbit polyclonal antibody (invitrogen, Waltham, MA, USA) for 50 min at room temperature at a 200:1 ratio. After washing with PBS, the slides were reacted with a secondary antibody (histo-fine simple-stain MAX-PO(R), Nichirei, Tokyo, Japan) for 30 min at room temperature. After coloration with DAB, the slides were rinsed with running water, and each was stained with Meyer hematoxylin. Coloring was conducted after washing with running water. After the sections were dehydrated with isopropyl alcohol, they were cleared and mounted.

The cross-sectional area (CSA) and fibrosis index were analyzed using a BZ-X700 BZ analyzer (Keyence, Japan). The fibrosis index was calculated as the red-stained muscle area divided by the CSA.

### 4.9. Real-Time Polymerase Chain Reaction (PCR)

The muscles stored in the RNA Protect^®^ Tissue Reagent were frozen in liquid nitrogen within one week and were then physically crushed using a Cryo-Press CP-100WP (Microtech Nichion, Chiba, Japan). Total RNA was extracted from the crushed soleus samples using ISOGEN II (NIPPON Gene, Osaka, Japan). cDNA was synthesized using reverse transcription with a ReverTra Ace^®^ qPCR RT Master Mix (TOYOBO, Osaka, Japan) according to the manufacturer’s instructions. Quantitative real-time PCR was performed using a Biosystem 7300 (Applied Biosystems, Carlsbad, CA, USA) with TaqMan Assay-on-Demand gene expression primer/probe sets (Applied Biosystems). Each 25 µL reaction mixture contained 2 µL of cDNA (100 ng) and 12.5 µL of TaqMan Gene Expression PCR Master Mix (TOYOBO, Osaka, Japan) for the target gene. The amplification protocol consisted of 40 cycles of denaturation at 95 °C for 15 s and annealing and extension at 60 °C for 1 min. Relative changes in gene expression were calculated according to the comparative C_t_ method. As an internal control, 18S-ribosomal RNA expression was analyzed. The primer sequences for real-time PCR were as follows:

18S-ribosomal RNA forward: 5′-ATGAGTCCACTTTAAATCCTTTAACGA-3′

18S-ribosomal RNA reverse: 5′-CTTTAATATACGCTATTGGAGCTGGAA-3′

18S-ribosomal RNA probe: 5′-(FAM)ATCCATTGGAGGCAAGTCTGGTGC(BHQ)-3′

p70S6K forward: 5′-AGAGCTTTTGGCTCGGAAG-3′

p70S6K reverse: 5′-GACTCACATCCTCTTCAGATTGC-3′

Atrogin-1 forward: 5′-GAAGACCGGCTACTGTGGAA-3′

Atrogin-1 reverse: 5′-ATCAATCGCTTGCGGATCT-3′

EGLN1 forward: 5′-CGACCTGATACGCCACTGT-3′

EGLN1 reverse: 5′-GTTCCATTGCCCGGATAAC-3′

HIF-1α forward: 5′-TTTTCAAGCAGTAGGAATTGGAA-3′

HIF-1α reverse: 5′-GTGATGTAGTAGCTGCATGATCG-3′

VEGF forward: 5′-GCAGCTTGAGTTAAACGAACG-3′

VEGF reverse: 5′-GGTTCCCGAAACCCTGAG-3′

### 4.10. Statistical Analysis 

All experimental data are presented as the mean ± standard deviation. A parametric one-way analysis of variance was used to examine statistical differences among groups. The Tukey–Kramer test was used to determine specific differences between groups if the results were significant. In all analyses, *p* < 0.05 was considered to indicate a statistically significant difference.

## 5. Conclusions

This is the first study to report that in an animal model of RA, sustained hypoxia-controlled arthritis and joint destruction occurred, while simultaneously preventing atrophy and fibrosis of slow-twitch muscles. The combination of hypoxia and moderate-intensity treadmill running further enhanced the preventive effects on muscle atrophy. This suggests the possibility of simultaneously controlling joint destruction and preventing sarcopenia in RA patients by suppressing arthritis with hypoxia and preventing muscle atrophy with moderate exercise therapy.

## Figures and Tables

**Figure 1 ijms-24-09761-f001:**
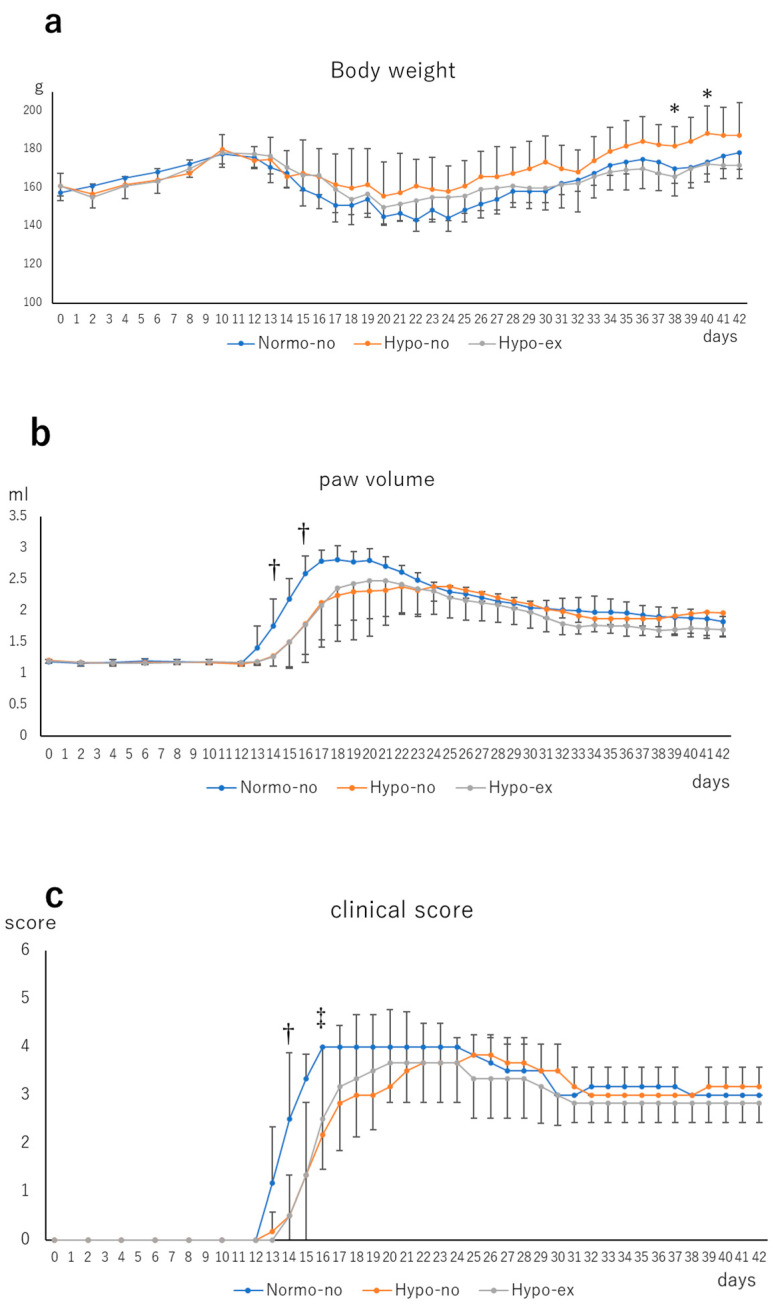
Temporal changes in body weight (**a**), paw volume (**b**), and clinical score (**c**) after immunization. All parameters were measured once every other day until day 12 and every day thereafter. * Hypo-no vs. Hypo-ex, *p* < 0.05; ^†^ Normo-no vs. Hypo-no, Hypo-ex, *p* < 0.05; ^‡^ Normo-no vs. Hypo-no, *p* < 0.05. n = 6 in each group.

**Figure 2 ijms-24-09761-f002:**
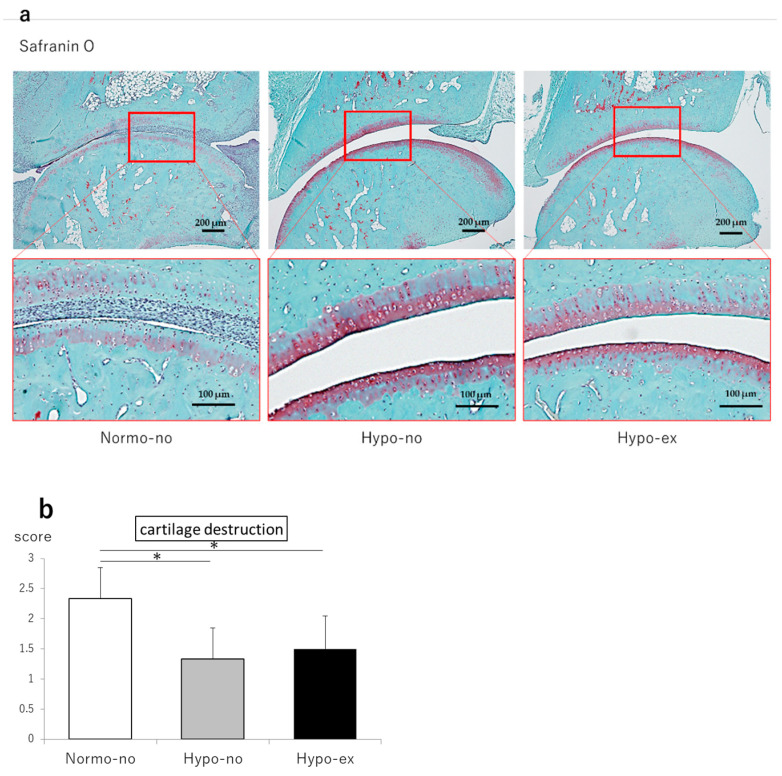
(**a**) Representative micrographs of safranin-O-stained sagittal sections of ankle joints. (**b**) The cartilage destruction scores based on the histological evaluation are shown. Each value represents the mean ± standard deviation. * *p* < 0.05. n = 6 in each group.

**Figure 3 ijms-24-09761-f003:**
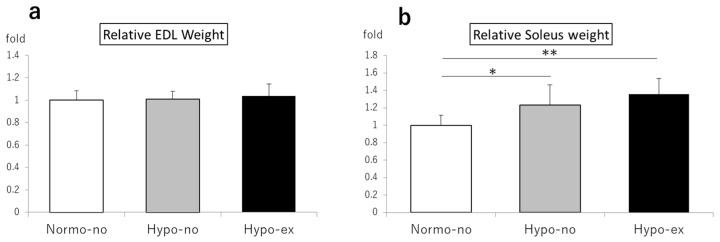
Relative weights of the EDL (**a**) and soleus (**b**) muscles were measured on day 44. The values of the muscle mass/BW ratio were normalized to the muscle mass/BW ratio of the Normo-no group. * *p* < 0.05, ** *p* < 0.01. BW, body weight; EDL, extensor digitorum longus. n = 6 in each group.

**Figure 4 ijms-24-09761-f004:**
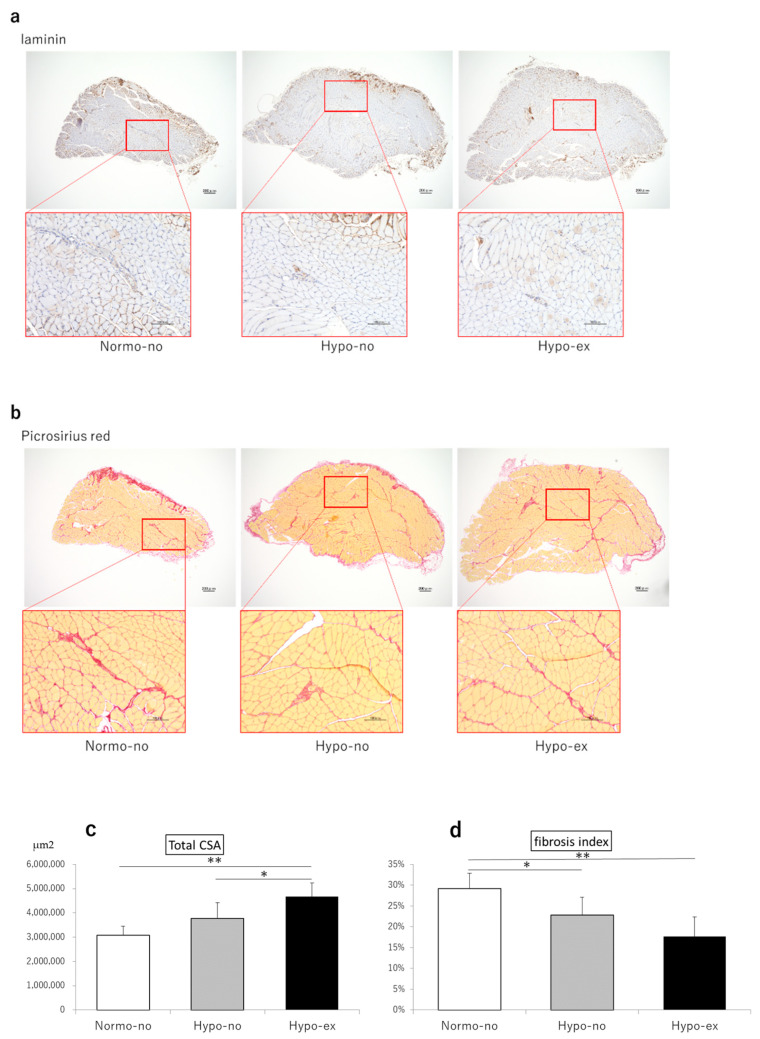
Representative microscopic images of laminin and picrosirius red-stained cross-sections (**a**,**b**) and histological analyses (**c**,**d**) of soleus muscles on day 44. The total CSA was measured on laminin-stained sections (**c**) and the fibrosis area was measured on picrosirius red-stained sections (**d**) of the soleus muscle. The fibrosis index is presented as fibrosis area/total CSA ratio. * *p* < 0.05, ** *p* < 0.01. Scale bars, 200 μm (low magnification) and 100 μm (high magnification). CSA, cross-sectional area. n = 6 in each group.

**Figure 5 ijms-24-09761-f005:**
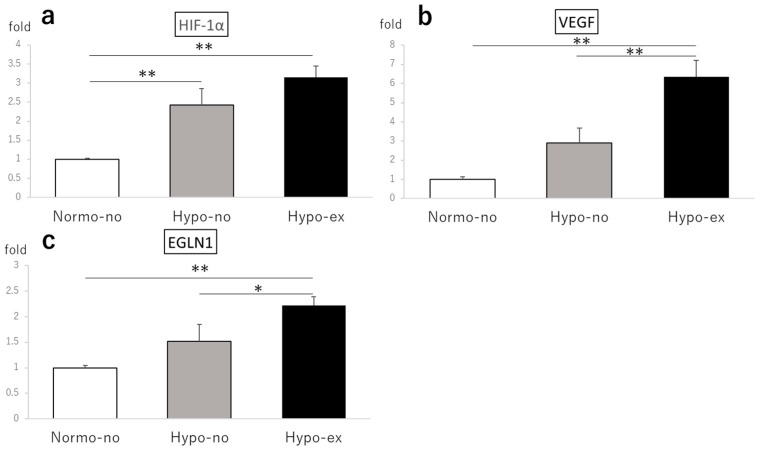
HIF-1α (**a**), VEGF (**b**), and EGLN1 (**c**) mRNA levels in the soleus muscle on day 2 were analyzed using a quantitative reverse-transcription polymerase chain reaction after a single bout of treadmill running. Each value represents the mean ± standard deviation. * *p* < 0.05, ** *p* < 0.01. EGLN1, egl-9 family hypoxia-inducible factor 1; HIF-1α, hypoxia-inducible factor-1α; VEGF, vascular endothelial growth factor. n = 4 in each group.

**Figure 6 ijms-24-09761-f006:**
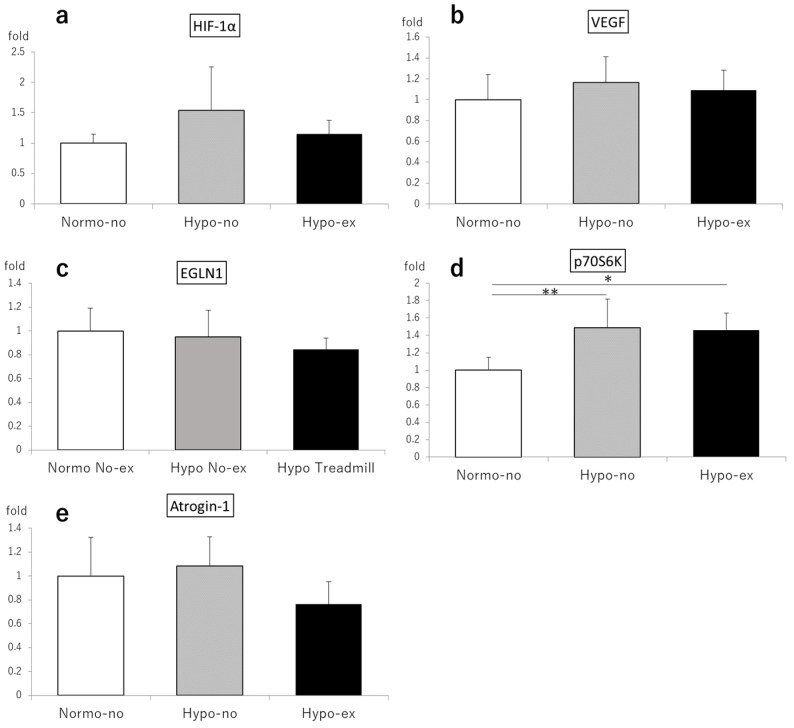
HIF-1α (**a**), VEGF (**b**), EGLN1 (**c**), p70S6K (**d**), and atrogin-1 (**e**) mRNA levels in soleus muscles on day 44 were analyzed using a quantitative reverse-transcription polymerase chain reaction. Each value represents the mean ± standard deviation. * *p* < 0.05, ** *p* < 0.01. EGLN1, egl-9 family hypoxia-inducible factor 1; HIF-1α, hypoxia-inducible factor-1α; VEGF, vascular endothelial growth factor. n = 6 in each group.

**Figure 7 ijms-24-09761-f007:**
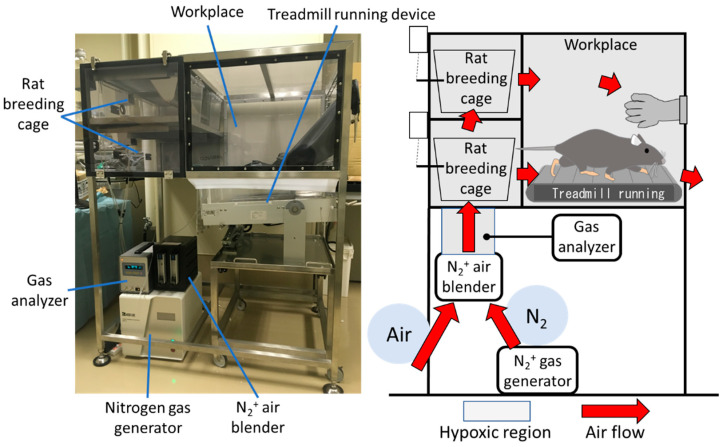
Hypoxic chamber. Nitrogen generated by the N_2_^+^ gas generator is mixed with outside air using an N_2_^+^ air blender to induce hypoxia. The oxygen concentration in the chamber can be adjusted as desired by circulating low oxygen levels through the chamber. Rats can rear and perform treadmill running seamlessly in this chamber.

**Figure 8 ijms-24-09761-f008:**
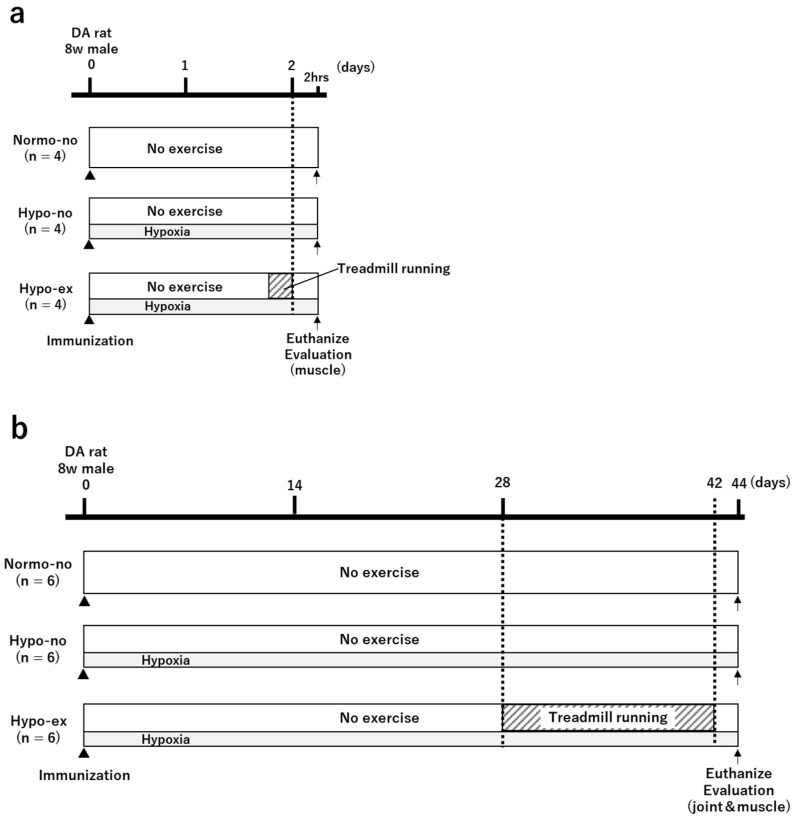
Experimental protocols. Eight-week-old male DA CIA rats were randomly divided into three groups: the Normo-no, Hypo-no, and Hypo-ex groups. Rats were euthanized 2 h after a single bout of treadmill running on day 2 (**a**) or 48 h after the final treadmill exercise on day 42 (**b**). Gene expression and histological analyses were performed to evaluate the effects of treadmill running on ankle joints, as well as fast- and slow-twitch muscles. CIA, collagen-induced arthritis; DA, Dark Agouti; Normo-no, normoxia no-exercise; Hypo-no, hypoxia no-exercise; Hypo-ex, hypoxia-exercise; w, week.

## Data Availability

Not applicable.

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
