# Peer review of "Hypoxia with or without Treadmill Exercises Affects Slow-Twitch Muscle Atrophy and Joint Destruction in a Rat Model of Rheumatoid Arthritis"

_ijms, 2023, doi:10.3390/ijms24119761_

Round 1
Reviewer 1 Report
The manuscript describes the effects of hypoxia and exercise associated with a hypoxic environment in a RA rat model. The authors showed that hypoxia controls the development of arthritis and prevents atrophy and fibrosis of slow-twitch muscles. The combination of hypoxia and treadmill training enhanced these effects.
Although the topic of this paper is interesting, the results only superficially describe the effects of treatment and do not examine the molecular mechanism behind these observations. The authors demonstrated that the hypoxic treatment condition increased RNA expression of HIF-1a and some of its target genes, which are already known to be activated in hypoxia. However, they added no evidence why hypoxia could be considered a valid approach to counteract skeletal muscle atrophy in a RA rat model. To validate this approach, the authors should perform the same experiment using a skeletal muscle knock-out HIF-1a rat. In addition, no results were shown regarding the effect of treatment on inflammation, which is critical for the development of RA.
Some other major concerns related to the manuscript:
- add the tiles in each scale bar in the graph. Fundamental.
-add images with higher magnification and brighter
- add the number of rats in the legends of each figure
-add the fold increase in the text
- Picuririus red is a stain suitable for measuring fibrosis, but not for calculating CSA. For this purpose, authors need to perform laminin staining, which marks the membrane of each fiber, and then they can calculate the CSA.
no specific comments
Author Response
Responses to comments by Reviewer 1
Point 1: Although the topic of this paper is interesting, the results only superficially describe the effects of treatment and do not examine the molecular mechanism behind these observations. The authors demonstrated that the hypoxic treatment condition increased RNA expression of HIF-1a and some of its target genes, which are already known to be activated in hypoxia. However, they added no evidence why hypoxia could be considered a valid approach to counteract skeletal muscle atrophy in a RA rat model. To validate this approach, the authors should perform the same experiment using a skeletal muscle knock-out HIF-1a rat. In addition, no results were shown regarding the effect of treatment on inflammation, which is critical for the development of RA.
Response 1: Thank you for this pertinent comment. You have raised an important question. As pointed out by Reviewer 1, measuring mRNA expression of hypoxia-inducible factor (HIF)-1α and some of its related genes is not sufficient. One of the study authors investigated the effects of HIF-1α on articular cartilage under mechanical stimulation and the associated mechanisms by knocking down HIF-1α (Shimomura, S.; Inoue, H.; Arai, Y.; Nakagawa, S.; Fujii, Y.; Kishida, T.; Shin-Ya, M.; Ichimaru, S.; Tsuchida, S.; Mazda, O.; et al. Mechanical stimulation of chondrocytes regulates HIF-1α under hypoxic conditions. Tissue Cell. 2021, 71, 101574. doi: 10.1016/j.tice.2021.101574). We have added these results to the introduction and discussion. As we have not been able to elucidate the entire pathway, we have added this as a limitation. The following is the corrected text.
Introduction
‘Shimomura et al. reported that HIF-1α regulates aggrecan and ADAM-TS5 expression in response to mechanical stimulation under hypoxia.’ (lines 60-62)
Discussion
‘Shimomura et al. reported that HIF-1α regulates aggrecan and ADAM-TS5 expression in response to mechanical stimulation under hypoxia and general mechanical stimulation in articular cartilage under hypoxia while controlling cartilage homeostasis.’ (lines 206-209)
‘Experiments with pharmacological or genetic target inhibition to investigate the involved pathways have not been performed.’ (lines 254-257)
Point 2: add the tiles in each scale bar in the graph. Fundamental.
add images with higher magnification and brighter
Response 2: Thank you for this suggestion. Accordingly, we have added scale bars to Figures 2a and 4a. Moreover, brighter images at higher magnification are now shown in Figures 2a and 4a,b.
Point 3: add the number of rats in the legends of each figure
Response 3: Thank you for raising this important point. The number of rats has been added to the legends of Figures 1-6.
Point 4: add the fold increase in the text
Response 4: We appreciate your suggestion and changed the text as follows:
Results
‘However, the relative soleus weight values in the Hypo-no and Hypo-ex groups were 1.23- and 1.36-fold higher than those in the Normo-no group (p<0.05 and p<0.01, respectively; Figure 3b).’ (lines 107-109)
‘The relative muscle cross-sectional area of the soleus in the Hypo-ex group was significantly increased compared to those in the Normo-no and Hypo-no groups (1.52- and 1.24-fold, p<0.01 and p<0.05, respectively; Figure 4b). The fibrosis rates of the soleus muscles in the Hypo-no and Hypo-ex groups were 22% and 40% decreased compared to that in the Normo-no group (p<0.05 and p<0.01, respectively; Figure 4c).’ (lines 118-123)
‘On day 2, the HIF-1α expression levels in the soleus muscles of the Hypo-no and Hypo-ex groups were 2.43- and 3.15-fold higher than those of the Normo-no group (p<0.01; Figure 5a). Likewise, the expression levels of vascular endothelial growth factor (VEGF) in the Hypo-ex group were elevated compared to those in the Normo-no and Hypo-no groups (6.35- and 2.19-fold, respectively, p<0.01; Figure 5b), as were the expression levels of egl-9 family hypoxia-inducible factor 1 (EGLN1) in the Hypo-ex group compared to those in the Normo-no and Hypo-no groups (2.22- and 1.47-fold, p<0.01 and p<0.05, respectively; Figure 5b,c).’(lines 137-144)
‘However, the expression of p70S6K, a marker of muscle protein synthesis, was significantly higher in the Hypo-no (1.49-fold) and Hypo-ex (1.46-fold) groups than in the Normo-no group (p<0.05 and p<0.01, respectively; Figure 6d).’ (lines 146-149)
Point 5: Picuririus red is a stain suitable for measuring fibrosis, but not for calculating CSA. For this purpose, authors need to perform laminin staining, which marks the membrane of each fiber, and then they can calculate the CSA.
Response 5: Thank you for providing these insights. We have included your thoughtful suggestion in the manuscript as follows:
Legend of Figure 4:
‘Representative microscopic images of laminin- and picrosirius red-stained cross-sections (a, b) and histological analyses (c, d) of soleus muscles on day 44. The total CSA was measured on laminin-stained sections (c), and the fibrosis area was measured on picrosirius red-stained sections (d) of the soleus muscle.’
Results
‘To histologically evaluate the muscles of each group, we performed laminin picrosirius-red staining of the soleus muscle (Figure 4a,b).’ (lines 117-118)
Materials and Methods
‘Laminin staining was using the following protocol: Slides were deparaffinized and hydrated. Slides were rinsed with distilled water, and treated by proteinase K for 8 min at room temperature. Slides were treated 3% hydrogen peroxide water and reacted with anti-laminin rabbit polyclonal antibody (invitrogen, Waltham, USA) for 50 min at room temperature at 200:1. After washing with PBS, the slides were reacted with a secondary antibody (histo-fine simple-stain MAX-PO(R), Nichirei, Tokyo, Japan) for 30 min at a room temperature. After coloration with DAB, the slides were rinsed with running water and each stained with Meyer hematoxylin. The coloring was done after washing with running water. After the sections were dehydrated with isopropyl alcohol, they were cleared and mounted.’ (lines 346-355)
Reviewer 2 Report
This is an interesting paper and I thoroughly enjoyed reviewing this manuscript. The paper is generally well written and structured. In particular, the authors investigated the beneficial effect of exercise therapy, in combination with hypoxic environment, in preventing joint destruction and muscle atrophy in rat models of RA. Overall, the manuscript is well-written, clear and sheds new light on potential beneficial effects of the combining treadmill running with a sustained hypoxic environment in preventing joint destruction, while slowing muscle atrophy and fibrosis in rats with collagen-induced arthritis (CIA).
I have only some requests and suggestions for revision, regarding a point which, in my opinion, is of paramount importance. The critical point is the definition of the level (intensity) of treadmill training. In fact, as reported in the current literature, distinct exercise training schedule of different intensity can cause a number of physiological adaptations in different organs, and especially at the level of adrenal gland, which represents an early target of physical exercise. In particular, evidence shows that the hypothalamus-pituitary-adrenocortical axis, as well as the sympatho-adrenomedullary system, is mainly involved in mediating the physiological effects of physical. Again, several morphological and biochemical changes were reported with reference to oxidative metabolism and muscle fiber composition in the mouse. In line with this, while slight to moderate levels seemed to be beneficial to cartilage health, more strenuous exercise may be detrimental. Thus, if exercise is thought to act like a drug preventing joint disease, the dosage may be critical to its success.
At present, it does not clearly emerge from the paper the characteristic and intensity of treadmill running protocol. Do the authors have provided an incremental exercise test? If not, please comment on this point. Therefore, in the opinion of the reviewer, it would be important that the Author implement the paper by discussing and citing previous studies, some key of them reported below, and clearly discuss all these points and compare their treadmill running protocol with previous literature to highlight whether it refers to a low-, moderate or high-intensity treadmill training. This would add significance to the paper. Also results, data interpretation and conclusion should be revised considering such points.
- Bartalucci A, Ferrucci M, Fulceri F, Lazzeri G, Lenzi P, Toti L, Serpiello FR, La Torre A, Gesi M. High-intensity exer- cise training produces morphological and biochemical changes in adrenal gland of mice. Histol Histopathol 27, 753–769, 2012.
- Toti, L.; Bartalucci, A.; Ferrucci, M.; Fulceri, F.; Lazzeri, G.; Lenzi, P.; Soldani, P.; Gobbi, P.; La Torre, A.; Gesi, M. High-intensity exercise training induces morphological and biochemical changes in skeletal muscles. Biol. Sport 2013, 30, 301–309
- Shimomura S, Inoue H, Arai Y, Nakagawa S, Fujii Y, Kishida T, Ichimaru S, Tsuchida S, Shirai T, Ikoma K, Mazda O, Kubo T. Treadmill Running Ameliorates Destruction of Articular Cartilage and Subchondral Bone, Not Only Synovitis, in a Rheumatoid Arthritis Rat Model. Int J Mol Sci. 2018 Jun 3;19(6):1653. doi: 10.3390/ijms19061653. PMID: 29865282; PMCID: PMC6032207.
- Zhou X, Cao H, Wang M, Zou J, Wu W. Moderate-intensity treadmill running relieves motion-induced post-traumatic osteoarthritis mice by up-regulating the expression of lncRNA H19. Biomed Eng Online. 2021 Nov 18;20(1):111. doi: 10.1186/s12938-021-00949-6. PMID: 34794451; PMCID: PMC8600697
- Fernandes MS, Sabino-Arias IT, Dionizio A, Fabricio MF, Trevizol JS, Martini T, Azevedo LB, Valentine RA, Maguire A, Zohoori FV, L Amaral S, Buzalaf MAR. Effect of Physical Exercise and Genetic Background on Glucose Homeostasis and Liver/Muscle Proteomes in Mice. Metabolites. 2022 Jan 25;12(2):117. doi: 10.3390/metabo12020117. PMID: 35208192; PMCID: PMC8878675.
- González-Chávez SA, López-Loeza SM, Acosta-Jiménez S, Cuevas-Martínez R, Pacheco-Silva C, Chaparro-Barrera E, Pacheco-Tena C. Low-Intensity Physical Exercise Decreases Inflammation and Joint Damage in the Preclinical Phase of a Rheumatoid Arthritis Murine Model. Biomolecules. 2023; 13(3):488. https://doi.org/10.3390/biom13030488
The English grammar and style are fine.
Author Response
Responses to comments by Reviewer 2
Point 1: This is an interesting paper and I thoroughly enjoyed reviewing this manuscript. The paper is generally well written and structured. In particular, the authors investigated the beneficial effect of exercise therapy, in combination with hypoxic environment, in preventing joint destruction and muscle atrophy in rat models of RA. Overall, the manuscript is well-written, clear and sheds new light on potential beneficial effects of the combining treadmill running with a sustained hypoxic environment in preventing joint destruction, while slowing muscle atrophy and fibrosis in rats with collagen-induced arthritis (CIA).
I have only some requests and suggestions for revision, regarding a point which, in my opinion, is of paramount importance. The critical point is the definition of the level (intensity) of treadmill training. In fact, as reported in the current literature, distinct exercise training schedule of different intensity can cause a number of physiological adaptations in different organs, and especially at the level of adrenal gland, which represents an early target of physical exercise. In particular, evidence shows that the hypothalamus-pituitary-adrenocortical axis, as well as the sympatho-adrenomedullary system, is mainly involved in mediating the physiological effects of physical. Again, several morphological and biochemical changes were reported with reference to oxidative metabolism and muscle fiber composition in the mouse. In line with this, while slight to moderate levels seemed to be beneficial to cartilage health, more strenuous exercise may be detrimental. Thus, if exercise is thought to act like a drug preventing joint disease, the dosage may be critical to its success.
At present, it does not clearly emerge from the paper the characteristic and intensity of treadmill running protocol. Do the authors have provided an incremental exercise test? If not, please comment on this point. Therefore, in the opinion of the reviewer, it would be important that the Author implement the paper by discussing and citing previous studies, some key of them reported below, and clearly discuss all these points and compare their treadmill running protocol with previous literature to highlight whether it refers to a low-, moderate or high-intensity treadmill training. This would add significance to the paper. Also results, data interpretation and conclusion should be revised considering such points.
- Bartalucci A, Ferrucci M, Fulceri F, Lazzeri G, Lenzi P, Toti L, Serpiello FR, La Torre A, Gesi M. High-intensity exer- cise training produces morphological and biochemical changes in adrenal gland of mice. Histol Histopathol 27, 753–769, 2012.
- Toti, L.; Bartalucci, A.; Ferrucci, M.; Fulceri, F.; Lazzeri, G.; Lenzi, P.; Soldani, P.; Gobbi, P.; La Torre, A.; Gesi, M. High-intensity exercise training induces morphological and biochemical changes in skeletal muscles. Biol. Sport 2013, 30, 301–309
- Shimomura S, Inoue H, Arai Y, Nakagawa S, Fujii Y, Kishida T, Ichimaru S, Tsuchida S, Shirai T, Ikoma K, Mazda O, Kubo T. Treadmill Running Ameliorates Destruction of Articular Cartilage and Subchondral Bone, Not Only Synovitis, in a Rheumatoid Arthritis Rat Model. Int J Mol Sci. 2018 Jun 3;19(6):1653. doi: 10.3390/ijms19061653. PMID: 29865282; PMCID: PMC6032207.
- Zhou X, Cao H, Wang M, Zou J, Wu W. Moderate-intensity treadmill running relieves motion-induced post-traumatic osteoarthritis mice by up-regulating the expression of lncRNA H19. Biomed Eng Online. 2021 Nov 18;20(1):111. doi: 10.1186/s12938-021-00949-6. PMID: 34794451; PMCID: PMC8600697
- Fernandes MS, Sabino-Arias IT, Dionizio A, Fabricio MF, Trevizol JS, Martini T, Azevedo LB, Valentine RA, Maguire A, Zohoori FV, L Amaral S, Buzalaf MAR. Effect of Physical Exercise and Genetic Background on Glucose Homeostasis and Liver/Muscle Proteomes in Mice. Metabolites. 2022 Jan 25;12(2):117. doi: 10.3390/metabo12020117. PMID: 35208192; PMCID: PMC8878675.
- González-Chávez SA, López-Loeza SM, Acosta-Jiménez S, Cuevas-Martínez R, Pacheco-Silva C, Chaparro-Barrera E, Pacheco-Tena C. Low-Intensity Physical Exercise Decreases Inflammation and Joint Damage in the Preclinical Phase of a Rheumatoid Arthritis Murine Model. Biomolecules. 2023; 13(3):488. https://doi.org/10.3390/biom13030488
Response 1: Thank you for your careful perusal of our manuscript. Based on your comments, we have cited the publications provided by you and corrected the Discussion. In this study, we did not perform incremental exercise tests. However, our group previously reported that moderate-intensity exercise has a positive effect on the articular cartilage in a rat model of rheumatoid arthritis (Shimomura, S.; Inoue, H.; Arai, Y.; Nakagawa, S.; Fujii, Y.; Kishida, T.; Ichimaru, S.; Tsuchida, S.; Shirai, T.; Ikoma, K.; et al. Treadmill running ameliorates destruction of articular cartilage and subchondral bone, not only synovitis, in a rheumatoid arthritis rat model. Int. J. Mol. Sci. 2018, 19, 1653. doi: 10.3390/ijms19061653). For this reason, the current experiment was limited to moderate-intensity exercise. As we recognize that low-intensity exercise and other exercise conditions may have a more positive effect on articular cartilage, we have added this point as a limitation.
Based on your valuable comments, we have changed the text as follows:
Discussion
‘Bartalucci et al. reported that the effects of training are dependent on complex, adaptive changes which are induced by acute physical exercise at different levels. In particular, the hypothalamus-pituitary-adrenocortical axis, as well as the sympatho-adrenomedullary system, are mainly involved in mediating the physiological effects of physical exercise [22]. Toti et al reported that high-intensity exercise, in addition to metabolic changes consisting of a decrease in blood lactate and body weight, induces an increase in mitochondrial enzyme levels and slow fiber numbers in different skeletal muscles of mice, which indicates an exercise-induced increase in aerobic metabolism [23]. Excessive mechanical stress on articular cartilage is a risk for osteoarthritis, whereas moderate mechanical stress is anabolic [24,25]. González-Chávez et al. reported that low-intensity physical exercise decreases joint damage and expression of RA-related genes and signaling pathways [26]. Shimomura et al. reported that HIF-1α regulates aggrecan and ADAM-TS5 expression in response to mechanical stimulation under hypoxia and general mechanical stimulation in articular cartilage under hypoxia while controlling cartilage homeostasis [17].’ (lines 195-209)
‘In the present study, the combined effects of 14-day moderate-intensity treadmill running and 42-day hypoxia on the suppression of joint swelling and destruction in CIA rats were investigated. The results indicate that the combination of sustained hypoxia and moderate-intensity treadmill exercise has the same inhibitory effect on joint destruction as sustained hypoxia alone.’ (lines 212-216)
‘Moreover, low-intensity exercise or other exercise conditions may have a more positive effect on articular cartilage and slow-twitch muscles.’ (lines 254-257)
Conclusions
‘The combination of hypoxia and moderate-intensity treadmill running further enhanced the preventive effects on muscle atrophy.’ (lines 397-398)
Round 2
Reviewer 1 Report
Dear authors,
Thank you for your responses.
Here are some more minor concerns:
The quality of the images and graphs is still very poor. I do not know if it's a technical problem or if the images are low resolution, but they are all grainy
line 125 insert "and" between Laminin/Picrosirius Red
Figure 4 shows two Picrosirius Red sequences
Add the title y-axis to all graphs. It is important to know the unit of measurement you are using.
no request
Author Response
Responses to comments by Reviewer 1
Point 1: The quality of the images and graphs is still very poor. I do not know if it's a technical problem or if the images are low resolution, but they are all grainy.
Response 1: Thank you for this suggestion. I have attempted to solve this problem by attaching the Figures separately from the text.
Point 2: line 125 insert "and" between Laminin/Picrosirius Red
Response 2: Thank you for raising this important point. I inserted “and” between Laminin/Picrosirius Red.
Point 3: Figure 4 shows two Picrosirius Red sequences
Response 3: We appreciate your suggestion. Figures have been prepared to make the relationship between low and high magnification in Figures easier to understand.
Point 4: Add the title y-axis to all graphs. It is important to know the unit of measurement you are using.
Response 4: Thank you for providing these insights. I added the title y-axis to all graphs.